# Utilization of insecticide-treated bed nets among pregnant women in Myanmar–analysis of the 2015–2016 Demographic and Health Survey

**Pyae Linn Aung** [1]**, Kyawt Mon Win**[2]**, Kyaw Lwin Show** [3]

**1** Myanmar Health Network Organization, Yangon, Myanmar, **2** Department of Public Health, Ministry of Health, Nay Pyi Taw, Myanmar, **3** Department of Medical Research, Ministry of Health, Yangon, Myanmar

☯ These authors contributed equally to this work.

* pyaelinnag@gmail.com

## Abstract

### Background

Due to the effectiveness of insecticide-treated nets (ITNs), most malaria-endemic countries resort to free distributions in the population with particular attention to pregnant women, a more vulnerable group. However, the mere issuance of ITNs does not usually translate to proper utilization. This study aimed to examine the utilization of ITNs and its associated factors among pregnant women in Myanmar.

### Methods

The data analyzed in this cross-sectional study were extracted from available survey datasets of the 2015–16 Myanmar Demographic Health Survey. The secondary data were presented using a chart, descriptive statistics and inferential statistics including simple and multiple logistic regression models. All analyses were performed using STATA, Version 15. A p-value <0.05 was considered statistically significant.

### Results

Of 466 currently pregnant women, the majority (96%) possessed bed nets for sleeping. Among them, 15.9% slept without a bed net the night before the survey, while 65.7% slept with untreated nets. Only about 1 in 5 (18.4%) slept under ITNs. In the multivariate logistic regression analysis, pregnant women residing in delta and lowland regions [adjusted odds ratio (aOR) = 7.70, 95% confidence interval (CI): 3.62, 16.38], plains (aOR = 7.09, 95%CI: 3.09, 16.25) or hilly areas (aOR = 4.26, 95%CI: 1.91, 9.52) were more likely to report non-utilization of ITNs than those residing in coastal regions.

### Conclusion

Relatively poor ITN utilization was observed among pregnant women in Myanmar. Health promotion activities for ITN utilization should be implemented especially for pregnant

**Data Availability Statement:** The whole dataset of the Myanmar Demographic Health Survey (MDHS) can be requested from the DHS website (https://

dhsprogram.com/). All relevant data of this study are within the manuscript.

**Funding:** The author(s) received no specific funding for this work.

**Competing interests:** The authors have declared that no competing interests exist.

women residing in the delta, lowland, plain and hilly regions. Other social-behavioral factors including perceived susceptibility to malaria, knowledge of ITNs, and attitude towards ITN that might favor the non-utilization of ITNs need to be further explored.

## Introduction

In 2020, the World Health Organization (WHO) reported almost 241 million malaria cases and 0.6 million deaths in the world [1]. Furthermore, 24 to 40% of the pregnant women in each region were exposed to malaria infection during pregnancy [1]. Pregnant women are threefold more likely to contract severe malaria and possess twice the risk of mortality than nonpregnant women [2]. Malaria in pregnancy is defined as a medical emergency and ought to be treated in hospital settings to prevent further adverse consequences, including maternal deaths [3]. In 2019, a total of 56,640 malaria cases and 14 malaria-related deaths were reported in Myanmar, and 365 cases involved pregnant women [1]. The malaria vulnerability of pregnant women would be a major impediment to achieving the countrywide malaria elimination target by 2030. Therefore, all pregnant women should always use good malaria preventive practices. However, targeted malaria interventions towards pregnant women remain limited except for the prioritized distribution of insecticide-treated nets (ITNs). ITNs are a form of personal protection that has been proved to prevent malaria infection, and reduce the risk of severe sickness, and mortality due to malaria [4].

Many studies have shown the possible effects of important malaria preventive practices in reducing the number of newly confirmed malaria cases and interrupting localized malaria transmission [5, 6]. The utilization of ITNs is internationally recognized as one of the most efficient and effective malaria preventive tools [7]. Thus, the distribution of insecticidal nets should happen regularly in every malaria-endemic country. In Myanmar, impregnation of conventional bed nets is not usually used unless an abnormal number of malaria cases occur in an area [8]. Instead, all malaria projects conducted mass distribution of insecticidal nets, either ITNs or long-lasting insecticide-treated nets (LLINs) (hereafter included under ITNs), every other year according to the National Strategic Plan. The topping-up replenishments covering malaria vulnerable groups, including forest-related workers, pregnant women, and children, were followed each year until the next cycle of mass distribution [8]. Nevertheless, proper utilization of ITNs among the community remained poor despite growing evidence of net ownership status. Consequently, with the financial support from donors including the Global Fund in Myanmar, about 11 million ITNs were distributed in 2019 [2]. Overall, as per the 2014 nationwide population census [9], the numbers of distributed ITNs were estimated to represent 50% of the total country population or 75% of the rural population. The reported utilization rates fluctuated by different population groups– 19.5% in the nationwide general population [10], 46.6% among pregnant women living in artemisinin-resistant areas [11] and 52% in migrant populations [12], respectively.

Entomologic surveys across the country have revealed the presence of abundant malaria vectors such as *Anopheles minimus* and *A. dirus* [13]. Thus, a need exists for immediate attention to strengthening the use of ITNs especially in rural areas where malaria is endemic and in other malaria vulnerable groups like pregnant women. Additionally, it remains critical to investigate the barriers to why people are not properly using the nets regardless of ownership status. Other studies in Thailand and Myanmar stated that types of occupation, sex, age, family wealth index, knowledge on malaria transmission, net ownership, household roofing, source

of family income and geographical zones were associated factors for the use or nonuse of ITNs among the general population as well as migrant workers [12, 14–17]. However, none of the studies represented nationwide samples. By understanding the sociodemographic factors related to the utilization of ITNs, more specific interventions such as targeted health education sessions could be formulated and initiated.

In Myanmar, the one and only Demographic Health Survey to date [18] was carried out during 2015 and 2016. The final report included much demographic and socio-economic data describing the common health problems of the citizens across the country. The data, including bed net utilization among pregnant women, were mainly presented in descriptive styles of analysis. However, according to the survey [18], the ITNs utilization rate in each state or region was below average. The lowest utilization rates have been observed in peripheral areas including Chin State (38.5%) followed by Kayah State (39.7%). The highest utilization areas were one state along the Thailand-Myanmar border–Karen State (80.9%) and an urban area– Yangon Region (76.2%). One other published paper [10] addressed ITNs utilization among under-5 children and reported poor utilization of ITNs as well as underlying factors. Here we further analyzed the nationwide data to explore the ownership and utilization of insecticidal nets and the related variables towards non-utilization of insecticidal bed nets among pregnant women in Myanmar. The results will be useful for formulating targeted health education interventions to boost bet net utilization for preventing malaria transmission among pregnant women in Myanmar and potentially other Greater Mekong Subregion countries.

## Methods

### Study design and source of data

This study comprised an analytical cross-sectional study using secondary data from the Myanmar Demographic and Health Survey 2015–16 (MDHS 2015–16).

### Myanmar Demographic and Health Survey 2015–16

To date, MDHS 2015–16 is the first and latest nationally representative survey to collect comprehensive data concerning basic demographic, socioeconomic, and health indicators of women and men aged 14 to 49 years residing in 15 states and regions. To obtain the representative samples for the whole country, the survey followed a stratified two-stage cluster sampling design yielding a response rate of 98%, ensuring the representativeness of the data. The first step was to estimate the numbers and points of clusters to be chosen either at the state or regional level and urban or rural areas. A total of 441 clusters have been included. This was followed by sampling a fixed number of 30 households from each cluster. All women aged 15 to 49 years in the selected households and all men aged 15 to 49 years in every second selected household were interviewed.

Three sets of questionnaires (households, men, and women) were used as a data collection tool. The contents were aligned with other worldwide DHS surveys except for some adjustments to the specific local context. The final questionnaire used for data collection was entirely in Burmese translated from English through various steps including a pretest. A training was organized for nine data collection masters recruited from the Ministry of Health. Those masters re-trained hundreds of field assistants from health departments, other non-governmental organizations, and ethnic groups. Data were collected from December 2015 to July 2016 using tablet computers. Data validation was also carried out at different levels. Field supervision together with technical monitoring visits were conducted by the DHS authority. A total of 466 currently pregnant women were included in the survey and were eligible to be involved in the present analysis.

## Data variables

The outcome variable of interest used in this study was currently pregnant women not sleeping under insecticide-treated nets (ITNs) the night before the survey. An ITN is defined as a factory-treated net that does not need to be treated again (long-lasting insecticide-treated net–LLIN), a pretreated net received during the last 12 months or a net that had been impregnated with insecticide within the last 9 months.

The independent predictor variables included were as follows: firstly completed age, counted as the age in years each participant responded during the interview. The ranges were categorized into 15 to 24 years, 25 to 34 years, and 35 to 49 years. Second, the highest education attained comprising the educational attainments of the participants grouped as no education, primary, secondary, or higher levels. Third, place of residence and region consisted of regions categorized according to their characteristics: delta and lowland (Ayeyawady, Yangon and Bago Regions, Mon, and the Karen States), hilly (Kachin, Kayah, Chin and Shan States), coastal (Rakhine State and Tanintharyi Region) and plains (Magway, Mandalay, Sagaing Regions, and Nay Pyi Taw Union Territory). Fourth, wealth indexes, based on economic status, income, and property ownership, were differentiated into richer, richest, middle, poor and poorest quintiles. Fifth, number of household members included the total numbers of household members categorized as <3, 4 to 7, and >7. Sixth, number of under 5 children in the household in three groups, i.e., none, one, or more than one. The seventh variable constituted duration of current pregnancy: the reported duration of pregnancy was coded as <5 months or 6 to 10 months. Finally, access to mass media exposure was defined as 'yes' when access could be gained to either one of television, newspaper, or radio at least once weekly.

## Data analysis

First, the utilization of insecticide-treated bed nets in the general population and among pregnant women was plotted using a bar chart. The percentages were calculated for pregnant women sleeping without nets, with untreated nets and with ITNs. To explore the true problem of ITN utilization status, utilization of ITN was estimated among total pregnant women and those with access to at least one ITN in their household. Second, general characteristics, demographic data, and socioeconomic status of pregnant women were analyzed using descriptive statistics including number, percentage, means and standard deviations. Last, the variables related to the non-utilization of insecticidal nets among pregnant women regardless of their ITN ownership status were explored using simple and multiple logistic regression models and presented using odds ratios with 95% confidence intervals. We included variables with a crude Chi-squared p-value of <0.2 in the multiple logistic regression model. All the analyses were performed using STATA (Version 15 STATA Corp., College Station, TX, USA). Weight factors and the 'svyset' command were applied to account for the two-stage stratified cluster sampling design. A p-value of <0.05 was considered statistically significant.

## Ethics consideration

The DHS had been conducted in Myanmar in line with ethical standards after obtaining the required approval from the Ministry of Health. The identities of all the respondents were kept confidential. The present study has comprised a secondary data analysis of its data and ethics approval was not required. The study title was also registered at the DHS program website. The authorization letter to use the survey datasets has been granted by the DHS program officials.

## Results

### Background characteristics of currently pregnant women included in the Myanmar Demographic and Health Survey 2015–16

Of 466 currently pregnant women, more than half were aged 25 to 34 years. Most attained at least primary level education. About 44% resided in the delta and lowland region followed by 25.4% in plains areas. A total of 359 (77%) were from rural areas, and nearly half (49.8%) were categorized in the two poorest quintiles. Many respondents (59.3%) possessed 4 to 7 members in their households. The studied women bore a pregnancy either for 1 to 5 or 6 to 10 months. Only a few (12.8%) had more than one under-five child in their households. Nearly 60% of pregnant women had exposure to mass media (television, newspaper, or radio) at least once weekly (**Table 1**).

### Bed net utilization among pregnant women involved in the Myanmar Demographic and Health Survey 2015–16

Most pregnant women, 96% (95% CI: 93.2, 97.7), possessed mosquito bed nets for sleeping while only 33% owned at least one insecticide-treated net (ITN) in their households (**Table 1**). However, 15.9% (95% CI: 12.4, 20.2) of pregnant women slept without a bed net the night before the survey. Many (65.7%) preferred to sleep under untreated nets. Only about one in five, 18.4% (95% CI: 14.7, 22.9) of total pregnant women slept under ITNs. Among pregnant women having access to at least one ITN in their households, almost 56% (86/154) slept under an ITN the night before the survey. Overall, about 84% of pregnant women slept in bed nets the night before the survey (**Fig 1**).

### Factors associated with non-utilization of insecticide-treated bed nets among currently pregnant women included in the Myanmar Demographic and Health Survey 2015–16

In the bivariate analysis, geographic location (p<0.001), place of residence (p = 0.019), wealth index (p = 0.034) and exposure to mass media (p = 0.014) were associated with non-utilization of ITNs. In the adjusted model analysis, the geographic location was significantly associated with the non-utilization of ITNs. Pregnant women who were residing in the delta and lowland region [adjusted Odd Ratio (aOR) = 7.70, 95% Confidence Interval (CI): 3.62, 16.38], plains (aOR = 7.09, 95%CI: 3.09, 16.25) and hills (aOR = 4.26, 95%CI: 1.91, 9.52) were more likely to practice non-utilization of ITNs than those living in coastal areas (**Table 2**).

## Discussion

The present study constitutes the first report documenting the utilization of insecticide-treated nets (ITNs) among the country's representative samples of pregnant women in Myanmar. The results estimated a relatively poor ITNs utilization among pregnant women who mostly preferred to use conventional or untreated nets. The result aligned with other research conducted in Myanmar and Cameroon, in which only 46.6% and 21.8% of pregnant women slept under ITNs [11, 19]. While Myanmar is struggling to attain its malaria elimination target within the given time frame amid limited resources availability [20], prevention becomes fundamental, and should be prioritized at least among malaria vulnerable groups like pregnant women. Every pregnant woman in Myanmar should seek antenatal care at the nearest or most convenient health center starting from the early first trimester followed by regular visits until delivery [21]. Healthcare officers could spread malaria-related health messages during each visit especially for pregnant women residing in ongoing malaria-endemic areas so that unnecessary

**Table 1. Background characteristics of currently pregnant women included in the Myanmar Demographic and Health Survey 2015–16 (n = 466).**

| Characteristics | | Number | Percentage |
|---|---|---|---|
| **Age** | | | |
| | 15–24 years | 131 | 28.2 |
| | 25–34 years | 249 | 53.3 |
| | 35–49 years | 86 | 18.5 |
| **Education** | | | |
| | No education | 63 | 13.6 |
| | Primary | 203 | 43.6 |
| | Secondary or higher | 200 | 42.8 |
| **Region** | | | |
| | Delta and lowland | 206 | 44.2 |
| | Hills | 92 | 19.8 |
| | Coastal | 50 | 10.6 |
| | Plains | 118 | 25.4 |
| **Residence** | | | |
| | Urban | 107 | 22.9 |
| | Rural | 359 | 77.1 |
| **Wealth index** | | | |
| | Poorest | 139 | 29.8 |
| | Poorer | 93 | 20.0 |
| | Middle | 76 | 16.4 |
| | Richer | 75 | 16.2 |
| | Richest | 82 | 17.6 |
| **Number of household members** | | | |
| | 1–3 | 123 | 26.4 |
| | 4–7 | 276 | 59.3 |
| | More than 7 | 67 | 14.3 |
| **Duration of current pregnancy** | | | |
| | 1–5 months | 240 | 51.4 |
| | 6–10 months | 226 | 48.6 |
| **Number of under 5 children in the household** | | | |
| | None | 219 | 47.0 |
| | One | 187 | 40.2 |
| | More than one | 60 | 12.8 |
| **Mass media exposure** | | | |
| | At least once a week | 279 | 59.8 |
| | Less than once a week | 187 | 40.2 |
| **Have mosquito net(s) for sleeping** | | | |
| | Yes | 447 | (96.0) |
| | No | 19 | (4.0) |
| **Have at least one insecticide treated net in household** | | | |
| | Yes | 154 | (33.0) |
| | No | 312 | (67.0) |

adverse consequences from the transmission of malaria could be eliminated. Whenever possible, observing proper ITNs utilization among pregnant women should also be accompanied by a home visit.

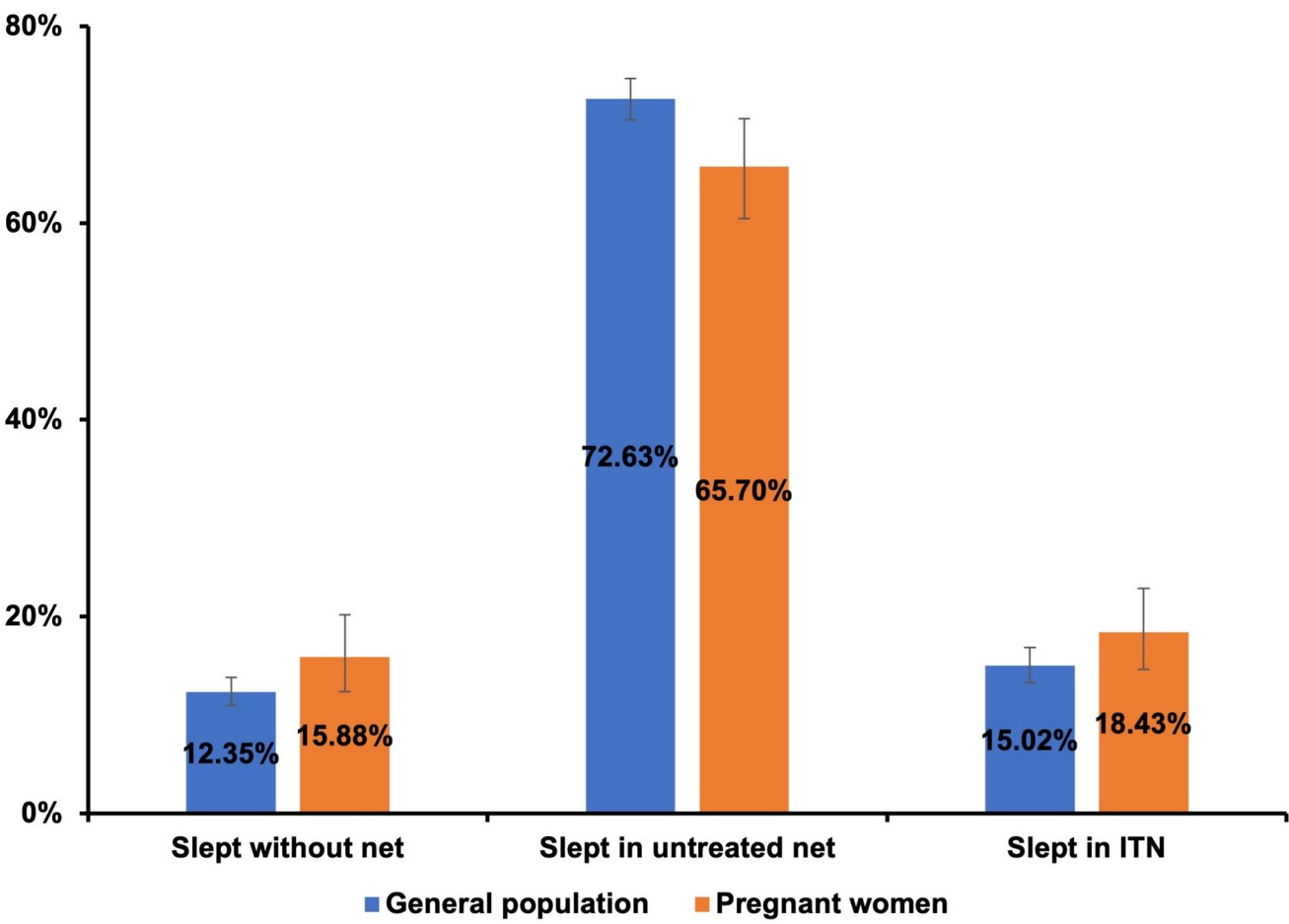

**Fig 1. Comparison of bed net utilization between the general population and pregnant women involved in the Myanmar Demographic and Health Survey 2015–16**

Given the higher malaria vulnerability among pregnant women, the prioritized distribution of insecticidal nets and expansion of their utilization should also be in place. In this study, although most pregnant women possessed either treated or untreated nets, some did not sleep under bed nets while only a few slept under ITNs. Related studies in Myanmar [11, 22] concluded the uniform results of poor utilization of ITNs despite an increase in distribution. The ITNs utilization rate might be influenced by socio-demographic factors such as age, sex, numbers of family members and type of net, including color and opacity [11, 16, 23, 24]. Some women preferred to use the kinds of nets that uphold utmost privacy rather than a transparent one [11]. Misbeliefs that insecticidal nets could be somehow harmful to their skin and give discomfort or hot feelings were reported among pregnant women [14, 19, 24]. Therefore, the national program should consider putting tremendous effort into distributing ITNs and continue expanding their use by reinforcing directions towards proper use.

Myanmar is composed of 14 states/ regions and one union territory. More than 70% of the total population lives in rural areas [18]. Presently, the country is focusing on flattening malaria caseloads in all states and regions. Particular attention and resource allocations are provided to these areas based on reported cases. Therefore, disparities might exist among each area in terms of strategy and malaria control activities including ITN distribution. Recent national data showed that Chin State presented the highest malaria burden area followed by

**Table 2. Factors associated with non-utilization of insecticide-treated bed nets among currently pregnant women included in the Myanmar Demographic and Health Survey 2015–16 (n = 466).**

| Characteristics | | Total | ITN non-utilization | | cOR | p-value | aOR | 95%CI |
|---|---|---|---|---|---|---|---|---|
| | | | n | (%) | | | | |
| **Total** | | 446 | 380 | 81.6 | | | | |
| **Age** | | | | | | 0.075 | | |
| | 15–24 years | 131 | 99 | 75.4 | Ref: | | Ref: | |
| | 25–34 years | 249 | 206 | 83.0 | 1.60 | | 1.63 | 0.91, 2.89 |
| | 35–49 years | 86 | 75 | 86.8 | 2.15 | | 2.02 | 0.91, 4.48 |
| **Education** | | | | | | 0.389 | | |
| | No education | 63 | 49 | 76.8 | Ref: | | - | - |
| | Primary | 203 | 164 | 80.6 | 1.25 | | - | - |
| | Secondary or higher | 200 | 168 | 84.1 | 1.59 | | - | - |
| **Region** | | | | | | <**0.001***  | | |
| | Delta and lowland | 206 | 183 | 89.0 | 10.03 | | 7.70 | **3.62, 16.38** |
| | Hills | 92 | 73 | 78.8 | 4.62 | | 4.26 | **1.91, 9.52** |
| | Coastal | 50 | 22 | 44.6 | Ref: | | Ref: | |
| | Plains | 118 | 102 | 86.3 | 7.84 | | 7.09 | **3.09, 16.25** |
| **Residence** | | | | | | **0.019*** | | |
| | Urban | 107 | 96 | 89.5 | 2.23 | | 1.20 | 0.53, 2.70 |
| | Rural | 359 | 284 | 79.2 | Ref: | | Ref: | |
| **Wealth index** | | | | | | **0.034*** | | |
| | Poorest | 139 | 110 | 79.2 | Ref: | | Ref: | |
| | Poorer | 93 | 70 | 75.3 | 0.80 | | 0.62 | 0.31, 1.27 |
| | Middle | 76 | 59 | 77.0 | 0.88 | | 0.60 | 0.27. 1.33 |
| | Richer | 75 | 66 | 87.8 | 1.89 | | 1.27 | 0.51, 3.18 |
| | Richest | 82 | 75 | 91.2 | 2.72 | | 1.41 | 0.48, 4.13 |
| **Number of household members** | | | | | | 0.588 | | |
| | 1–3 | 123 | 100 | 81.3 | Ref: | | - | - |
| | 4–7 | 276 | 229 | 82.7 | 1.10 | | - | - |
| | more than 7 | 67 | 51 | 77.3 | 0.78 | | - | - |
| **Duration of current pregnancy** | | | | | | 0.474 | | |
| | 1–5 months | 240 | 198 | 82.8 | Ref: | | - | - |
| | 6–10 months | 226 | 182 | 80.3 | 0.84 | | - | - |
| **Number of under 5 children in the household** | | | | | | 0.084 | | |
| | None | 219 | 186 | 84.9 | Ref: | | Ref: | |
| | One | 187 | 151 | 80.6 | 0.74 | | 0.86 | 0.48, 1.53 |
| | More than one | 60 | 43 | 72.5 | 0.47 | | 0.59 | 0.27, 1.26 |
| **Mass media exposure** | | | | | | **0.014*** | | |
| | At least once a week | 279 | 238 | 85.2 | Ref: | | Ref: | |
| | Less than once a week | 187 | 142 | 76.1 | 1.55 | | 0.91 | 0.51, 1.61 |

ITN: Insecticide-treated Net; cOR: Crude Odd Ratio; aOR: Adjusted Odd Ratio; CI: Confidence Interval; *Significance at p<0.05.

Karen State. Generally, a greater disease control effort delivered a larger outcome or improvement. However, this study observed the highest non-utilization status among pregnant women residing in delta and lowland regions (Ayeyawady, Yangon, and Bago Regions, Mon, and Karen States) than those living in coastal areas. All the areas except Karen State are in central Myanmar where malaria cases are steeply decreasing, and malaria elimination-specific

activities have been implemented since 2017 [8]. Once the malaria burden is reduced, people's perception of disease severity decreases [25]. The presence of ongoing political instabilities, conflicts, and the influx of population migration inside Karen State might hinder the proper ITNs utilization among pregnant women. Our overall results have been given weight by other studies [10, 17] in Myanmar in which people living in delta and lowland areas presented the poorest utilization status of ITNs.

Generally, malaria is more prevalent in rural areas where high vector density is located. Thus, rural communities might be more aware of the disease, and keep practicing preventive measures, than those living in urban areas. Consequently, pregnant women from the urban areas showed lesser ITN utilization status in this study. Other studies [4, 10] in Myanmar and Ghana also reported that people living in rural areas possessed a higher bed net utilization rate. Thus, malaria awareness-raising campaigns should be implemented primarily in urban areas of the delta, lowland, plain and hilly regions. Given that preventing re-establishment by inter-rupting onward transmission from the imported malaria cases is essential, pregnant women residing in urban areas should always follow good preventive practices.

Many studies concluded that the individual's economic status was associated with levels of malaria preventive practice in the community [10, 26–28]. Mostly, ITNs are distributed free of charge directly to the hands of pregnant women. Meanwhile, the country's GDP is at the lowest level. Universal Health Coverage is still in the skilling up phase [29]. Rural families with financial hardship are trying hard to earn a decent income for their daily survival and health-care seeking. Therefore, most are usually remote from routine malaria control interventions including awareness-raising campaigns. As per local culture, additionally, women always give priority to their children or other family members. When ITN ownership status is not high, pregnant women cannot utilize the ITN although they possess good malaria preventive knowledge. Need-based targeted interventions are recommended to take place among underprivileged individuals with low economic status in need of support. All families should possess at least one ITN per two persons in their households.

Disseminating health messages using mass media proved effective in many areas, including improving malaria preventive practice [23, 30]. Yet, the approach is likely to be more beneficial for remote areas with transportation difficulties and when in-person mass gathering activities are not allowed for the sake of controlling disease transmission, e.g., COVID-19 [31]. In Myan-mar, malaria-related health messages have been broadcast by the government's ministries and projects [32]. Therefore, no doubt pregnant women residing in areas with access to mass media should have noticed those health messages and know how to practice excellent bed nets utilization.

The study encountered a few limitations. First, it involved secondary data analysis of available data from countrywide demographic and health surveys. Therefore, the number of variables was limited. Other social-behavioral factors including misbeliefs, knowledge of ITN, attitude towards ITN and perceived susceptibility to malaria of pregnant women that might favor the non-utilization of ITNs need to be further explored. Second, the data obtained only by the questionnaire might under- or overestimate the actual situation. Respondents might also wrongly report between conventional bed nets and ITNs. However, these might have been escalated by data collection visits conducted during the DHS by observing ITN utilization. Third, the DHS survey was conducted from 2015 to 2016. Therefore, ITN ownership status might differ since then due to rigorous distributions made after the country's endorsement towards nationwide malaria elimination in 2015. Despite these limitations, this study provides essential information which could be relied upon to improve the effective utilization of ITNs especially among pregnant women and perhaps children under five; since they are vulnerable to malaria.

## Conclusions

Although the bed net ownership status was high, a relatively poor ITN utilization rate was observed among pregnant women in Myanmar. It may jeopardize the attainment of country-wide malaria elimination by 2030. Eliminating unwarranted maternal mortality due to malaria transmission and enhancing the utilization of ITNs among pregnant women is essential. Targeted health education activities among pregnant women should be implemented throughout the country. Next, healthcare professionals should implement routine health education sessions or awareness-raising campaigns for pregnant women during antenatal care visits to foster malaria preventive practices. The priority group includes vulnerable pregnant women residing in urban areas of the delta, lowland, plain and hilly regions.

## Acknowledgments

The authors are thankful to the Demographic and Health Surveys (DHS) Program for granting access to the survey datasets.

## Author Contributions

**Conceptualization:** Pyae Linn Aung, Kyawt Mon Win, Kyaw Lwin Show.

**Data curation:** Pyae Linn Aung, Kyaw Lwin Show.

**Formal analysis:** Kyaw Lwin Show.

**Methodology:** Pyae Linn Aung, Kyaw Lwin Show.

**Writing – original draft:** Pyae Linn Aung, Kyaw Lwin Show.

**Writing – review & editing:** Pyae Linn Aung, Kyawt Mon Win, Kyaw Lwin Show.

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
