## [Decision Letter · Decision Letter 0]

31 Jan 2022

PONE-D-22-00065Utilization of insecticide-treated bed nets to prevent malaria transmission among pregnant women in Myanmar – A secondary data analysis of the 2015-2016 nationwide Demographic Health SurveyPLOS ONE

Dear Dr. Aung,

Thank you for submitting your manuscript to PLOS ONE. After careful consideration, we feel that it has merit but does not fully meet PLOS ONE’s publication criteria as it currently stands. Therefore, we invite you to submit a revised version of the manuscript that addresses the points raised during the review process.

We look forward to receiving your revised manuscript.

Kind regards,

Myat Htut Nyunt, MMedSc

Academic Editor

PLOS ONE

Journal Requirements:

2. We note that Figure 1 in your submission contain copyrighted images. All PLOS content is published under the Creative Commons Attribution License (CC BY 4.0), which means that the manuscript, images, and Supporting Information files will be freely available online, and any third party is permitted to access, download, copy, distribute, and use these materials in any way, even commercially, with proper attribution. For more information, see our copyright guidelines: http://journals.plos.org/plosone/s/licenses-and-copyright.

Reviewers' comments:

Reviewer's Responses to Questions

**Comments to the Author**

1. Is the manuscript technically sound, and do the data support the conclusions?

Reviewer #1: Yes

Reviewer #2: Partly

Reviewer #3: Yes

2. Has the statistical analysis been performed appropriately and rigorously? 

Reviewer #1: No

Reviewer #2: Yes

Reviewer #3: Yes

3. Have the authors made all data underlying the findings in their manuscript fully available?

Reviewer #1: Yes

Reviewer #2: Yes

Reviewer #3: Yes

4. Is the manuscript presented in an intelligible fashion and written in standard English?

Reviewer #1: Yes

Reviewer #2: Yes

Reviewer #3: Yes

5. Review Comments to the Author

Reviewer #1: Thank you very much for this opportunity to revise the manuscript titled "Utilization of insecticide-treated bed nets to prevent malaria transmission among pregnant women in Myanmar – A secondary data analysis of the 2015-2016 nationwide Demographic Health Survey" that was submitted to PLOS ONE. The study is original research that utilized secondary data from 2015-2016 nationwide Demographic Health Survey-Myanmar. The study revealed very low rate of insecticide-treated bed nets usage among pregnant women. Insecticide-treated bed nets are the simplest way of preventing mosquito bites. Mosquito bites causes malaria. Malaria infection affects all persons in malaria endemic areas, however pregnant women and children under five are more vulnerable. In pregnancy, Malaria can cause anaemia, which lead to stillbirths and in some instance maternal morbidity and mortality. Due to the effectiveness of ITNs, most malaria endemic countries resort to free distributions to the population with special attention to the expectant mothers and postnatal mothers. However, in most countries the mere distribution of ITNs do not translate to utilisation. From above therefore, this study therefore is considered very important for policy directions. However, for this work to be published in Plos One, these are some comments for the authors in order to improve the manuscript:

Abstract

Methods under Abstract ought to be improved.

• Indicate the version of STATA used for the analysis.

• The study design in missing in this section.

• At what point, is a variable said to be significant in the inferential statistics?

Introduction

-The authors begin talking about Insecticides Treated bed Nets. However, in many European countries, the use of this method of malaria prevention is little known, so it could be interesting to add a sentence explaining what it consists of and what health benefits it has. This will help readers form countries in which malaria is not endemic to appreciate the importance of this study.

The authors may refer to a study in Ghana for the definition and benefits of ITNs … Utilization of Insecticides Treated Mosquito Nets (ITNs) Among Pregnant Women in Kassena-Nankana Municipality… Kindly refer to the introduction and cite appropriately.

-Kindly read the following published work in Myanmar and update line 78 to 81…. Other studies in Thailand and Myanmar stated that types of occupation, gender, age, socioeconomic status, net ownership, and residing places ….

Liu, H., Xu, J. W., Guo, X. R., Havumaki, J., Lin, Y. X., Yu, G. C., & Zhou, D. L. (2015). Coverage, use and maintenance of bed nets and related influence factors in Kachin Special Region II, northeastern Myanmar. Malaria journal, 14(1), 1-12.

Cheng, B., Htoo, S. N., Mhote, N. P. P., & Davison, C. M. (2021). Association between biological sex and insecticide-treated net use among household members in ethnic minority and internally displaced populations in eastern Myanmar. Plos one, 16(6), e0252896.

Methods

-Kindly specify if the study design is descriptive or analytical cross-sectional study

-Line 126 to 138 including the figure 1 should be moved to introduction.

- Kindly make a sub heading for dependent (outcome) and independent (explanatory) variables and provide vivid description to each.

- The criteria for inclusion into the logistics was not stated. This has the tendency affect the goodness of fit of the model.

See the following (concentrate on the data analysis only);

Mekonnen, B. D., & Wubneh, C. A. (2020). Prevalence and associated factors of contraceptive discontinuation among reproductive-age women in Ethiopia: using 2016 Nationwide Survey Data. Reproductive Health, 17(1), 1-10.

Abubakari, A., Taabia, F. Z., & Ali, Z. (2019). Maternal determinants of low birth weight and neonatal asphyxia in the Upper West region of Ghana. Midwifery, 73, 1-7.

Mutaru, A. M., Asumah, M. N., Ibrahim, M., Sumaila, I., Hallidu, M., Mbemah, J. M., ... & Zakaria, D. Y. (2021). Knowledge on Sexually Transmitted Infections (STIs) and sexual practices among Nursing Trainees in Yendi Municipality, Northern Region of Ghana. European Journal of Health Sciences, 6(4), 33-47.

Results

-Table 1; kindly check the percentages of Wealth index, it is more than 100.0%. Kindly check the values at 1 decimal place again and make necessary changes.

-Line 194, the authors discovered that 96% of the pregnant women possessed bed nets. However, in figure 2 no such information is captured. kindly reconcile the descriptions and figure 2.

-For consistency in reporting of findings …I suggest the authors reports findings as … 96% (95% CI: 93.2, 97.7)

-line 195-198, it is indicated that… “However, 16% (95% CI: 12.4, 20.2) of pregnant women slept without a bed net last night prior to the survey which was a bit higher than the general population (12%). Many of both groups preferred to sleep under the untreated nets. Only about one in five (18.4%, 95% CI: 14.7, 22.9) pregnant women slept under insecticide-treated bed nets”.

-From above, my understanding is that about 84% and 88% of pregnant women and general population respectively slept in bed nets a night to the survey. Is that correct?

-If above question is correct, how then did we get only 18.4% of pregnant women reported to be using insecticide-treated bed nets?

_How do we know that the other nets were not treated? In the introduction, I came across free distribution of bed nets aimed at meeting the 2030 targets by the country. It is the case that, majority of the 96% ownership of bed nets are not treated? Kindly explain and make it clear to the reader.

-The authors further indicated that, only 18.4% slept in ITNs. this is problematic. Because the dependent (outcome) variable was not clearly defined, it is becoming very difficult to understand these findings.

-I would suggest that the methods be clarified to give true meaning to these findings.

-Bed nets and insecticide-treated bed nets should not be used interchangeably. Insecticide-treated bed nets are bed nets, but not all bed nets are insecticide-treated bed nets.

-I also suggest definition of terms at the methods to give true meaning to the findings.

_Line 207 -215 is not very necessary. I suggest to the authors to expunged expunge it.

-Only one variable is significant at the adjusted model because the authors did not control confounding variables. I would suggest that, only significant variables or variables with a p value less than 0.2 at bivariate analysis level be considered in the adjusted model analysis. This should be clearly stated at the methods. You may refer to the materials provided at the methods sections.

Discussions

-The discussion is okay and up to date, However, if authors would not mind, I would suggest they add some studies from Africa to support the work. I am suggesting very recent studies for your perusal and update. Africa is another malaria endemic area and almost all the interventions outline in the study are being done in most Africa countries.

Ngouakam, H., Fru-Cho, J., & Tientche, B. (2021). Awareness, Use, Care of Insecticide-treated Bed Nets among Pregnant Women in Buea (Buea) and Bonassama (Douala). Health Science Journal, 15(2), 1-8.

Asumah, M. N., Akugri, F. A., Akanlu, P., Taapena, A., & Boateng, F. (2021). Utilization of insecticides treated mosquito bed nets among pregnant women in Kassena-Nankana East municipality in the upper east region of Ghana. Public Health Toxicology, 1(2), 1-11. (Read to see why rural folks tend to use ITNs more than the urban folks, and the wealth index too at the discussion section)

Tassembedo, M., Coulibaly, S., & Ouedraogo, B. (2021). Factors associated with the use of insecticide-treated nets: analysis of the 2018 Burkina Faso Malaria Indicator Survey. Malaria Journal, 20(1), 1-9.

-Limitation of this study should be mentioned at the end of the discussions.

Reviewer #2: General comments

The overall structure of the manuscript is good and written based on the commonly agreed structure. The discussion and conclusion section needs major revision based on the objective and the result of the study.

Specific comments

1. Title: it can be modified “utilization of insecticide treated bed nets among pregnant women in Myanmar: analysis of 2015-2016 demographic and health survey”. Because no need of mentioning ---to prevent malaria transmission… as it is known.

2. Abstract

2.1 Background- it lacks information why the authors did the analysis.

2.2 Result- The description of the use of bed net is not clear. It stated that 16% slept without bed net and 18.4 slept under ITN. What about others…16% +18.4%=34.4%, 65.6% are they users or not. Make it clear for the reader. In the main document, it is clear.

Mentioning the result of bivariate analysis in the abstract is not important.

2.3 Conclusion- the recommendation sentence is not based on the significant variable. Mass media exposure do not show a significant association with utilization.

Keywords: it is good if arranged alphabetically

3. Introduction

• Reference number one is outdated you can update and use the 2021 malaria report which indicates 241 million cases of malaria. (https://www.who.int/news-room/fact-sheets/detail/malaria)

• Based on line 51 and 52 sentence, malaria is not a problem of pregnant women. How do the authors explain the public health importance of this research for Myanmar?

• It is good if the authors indicate the interventions by different stakeholders to increase ownership and utilization of ITN among pregnant women.

4. Method

• It is good if line number 126 to 138 presented under the introduction section, which shows the intervention by different stakeholders.

5. Result

• Table 1- the bracket under percentage column for all percent is not necessary as it is written in a separate column. In the table the frequency added to 99.9% not 100%, check it.

• Line 196 seems discussion, not description of the result. It can be stated in the discussion section.

• Line 207-215 should be under bed net utilization, not the factors associated with bed net utilization.

6. Discussion

• The discussion should be guided by the objective of the study. However, most of the parts of the discussion is based on the general understanding not based on the result of the study. Only paragraph 2 and 3 focused discussion based on the objective.

• It lacks the limitation of the study. Many variables may be missed because it is a secondary data such as knowledge, attitude, risk perception etc.

7. Conclusion

• The recommendation part (line 310-318) is a general recommendation not based on the result of the study. Mass media and other intervention suggested is not the significant variables, which is associated with utilization of ITN in this study.

Reviewer #3: The authors focused on the ownership and utilisation insecticide treated nets among pregnant women in Myanmar. Women constitute with children, a major risk group and known to be vulnerable to the disease. Despite great achievement in malaria control during the past decade, due to various interventions, the disease remains a huge public concern in developing countries. This is due partly to low utilisation of ITNs and poor attitude to preventive measures. Therefore, this paper presented aimed at to understand factors associated with the non-utilization of ITNs among pregnant women in Myanmar. The manuscript revealed a relatively poor ITNs utilization rate pregnant woman in Myanmar which could jeopardise the attainment of the malaria elimination goal by 2030. Residing in plains and hilly areas were preditors of non-utilisation of ITNs among pregnant women in Myanmar. With regards to weakness, minor English revision may be useful. The manuscript was well written following PLOS ONE recommended outline. The manuscript represents a high level of scientific work and appear to be an interested topic for scientists works in the areas of ITNs universal coverage.

6. PLOS authors have the option to publish the peer review history of their article (what does this mean?). If published, this will include your full peer review and any attached files.

Reviewer #1: No

Reviewer #2: **Yes: **Teklemariam Gultie

Reviewer #3: **Yes: **bonaventure tientche

---

## [Author Response · Author response to Decision Letter 0]

3 Feb 2022

Dear Editors and Reviewers.

Thank you for the constructive comments, critiques, and valuable suggestions. We have now attempted to address each point and provide a point-by-point response in bold blue text in the Response to Reviewers file. Page and Line numbers are referred to clean manuscript file. Figure 1 that is similar to a copyright figure has now been excluded.

---

## [Decision Letter · Decision Letter 1]

21 Feb 2022

PONE-D-22-00065R1Utilization of insecticide-treated bed nets among pregnant women in Myanmar – analysis of the 2015-2016 Demographic and Health SurveyPLOS ONE

Dear Dr. Aung,

Thank you for submitting your manuscript to PLOS ONE. After careful consideration, we feel that it has merit but does not fully meet PLOS ONE’s publication criteria as it currently stands. Therefore, we invite you to submit a revised version of the manuscript that addresses the points raised during the review process.

We look forward to receiving your revised manuscript.

Kind regards,

Myat Htut Nyunt, MMedSc

Academic Editor

PLOS ONE

Journal Requirements:

Reviewers' comments:

Reviewer's Responses to Questions

**Comments to the Author**

1. If the authors have adequately addressed your comments raised in a previous round of review and you feel that this manuscript is now acceptable for publication, you may indicate that here to bypass the “Comments to the Author” section, enter your conflict of interest statement in the “Confidential to Editor” section, and submit your "Accept" recommendation.

Reviewer #1: (No Response)

Reviewer #2: All comments have been addressed

2. Is the manuscript technically sound, and do the data support the conclusions?

Reviewer #1: Yes

Reviewer #2: Yes

3. Has the statistical analysis been performed appropriately and rigorously? 

Reviewer #1: Yes

Reviewer #2: Yes

4. Have the authors made all data underlying the findings in their manuscript fully available?

Reviewer #1: Yes

Reviewer #2: Yes

5. Is the manuscript presented in an intelligible fashion and written in standard English?

Reviewer #1: Yes

Reviewer #2: Yes

6. Review Comments to the Author

Reviewer #1: Thank you very much for this opportunity to revise the manuscript titled "Utilization of insecticide-treated bed nets to prevent malaria transmission among pregnant women in Myanmar – A secondary data analysis of the 2015-2016 nationwide Demographic Health Survey" that was submitted to PLOS ONE. The study is original research that utilized secondary data from 2015-2016 nationwide Demographic Health Survey-Myanmar.

Abstract

* The secondary data 25 were “analyzed” using a chart, … Please replace analysed to “presented”

* Among them, 16% slept without a bed net the night before the survey, while more 30 than 65% slept with untreated nets. Only about 1 in 5 (18.4%) slept under ITNs. (16+65+18.4=99.4%) where is the remaining 0.6%.

*Introduction and methods are sound in my view.

Results

*In Table 1, please reference to the last two variables. The study reported that 96% of the pregnant have or owned or possessed a bed net. Yet only 33.0% were reported to have had ITNs. Please, are you sure these subjects could differentiate between bed nets and ITNs? You may have to have something around this in your limitation if you intend to leave it as captured now.

*In line 206 to 208, results on the general population are presented. Please, the general population is not your sample units. I think we should not report on that.

*In table 2, the second variables which is level of education has a p value of (0.389) at the bivariate level (COR). Given that you have set p<0.2 as the inclusion criteria into the adjusted model, it ought not to be part of the variables entered into the multivariate analysis (AOR). Kindly take it out.

*Kindly bold the significant variables.

*Kindly add something about the importance of this study just after the limitations. For examples, “… Despite these limitations, this study provides essential information which could be relied upon to improved upon the effective utilisation of ITNs especially among the pregnant women and perhaps children under five; since they are vulnerable to malaria…” Feel free to rephrase to suit your work.

Reviewer #2: General comments

The authors’ addressed most of my comments and made significant improvements of the manuscript. For further improvement of the manuscript, here under I suggest few modifications.

Specific comments

Abstract

Background- (Line 20-22) - the last paragraph that states the objective has to be rephrased. My suggestion “this study aimed to examine the utilization of ITN and associated factors among pregnant women in Myanmar”. Because utilization can be yes or no or utilized and non-utilized.

Conclusion- (Line 36-38) - in the recommendation part, still I am not convinced the suggestion of mass media. First, mass media exposure did not show any relation with utilization or non-utilization of ITN. Second, there is no evidence in this study whether the mentioned regions lack access to media. I think it is preferable to recommend to the government or any concerned organizations to give more emphasis in those regions to increase the level of ITN utilization. Probably suggesting further study to identify the factors that is not addressed by this study. There are many important factors missed in this research such as knowledge of ITN, attitude towards ITN, perceived susceptibility to malaria, etc.

Discussion

Line 284-294- is all about the general population, not related with pregnant women, which is the study population for this research. If I am not mistaken, are pregnant women involved in Gold mining and rubber tapping? If they are involved, it is acceptable explanation. May be you can check the reason for not utilizing is due to priority given to their children, if there is a culture of priority given to children in your country, etc. than discussing the reason of the general population.

Conclusion

See my comments in the conclusion section of the abstract.

7. PLOS authors have the option to publish the peer review history of their article (what does this mean?). If published, this will include your full peer review and any attached files.

Reviewer #1: **Yes: **Mubarick Nungbaso Asumah

Reviewer #2: **Yes: **Teklemariam Gultie

---

## [Author Response · Author response to Decision Letter 1]

24 Feb 2022

Dear Editors and Reviewers.

Thank you for the constructive comments, critiques, and suggestions. We have now attempted to address each point and provide a point-by-point response in bold blue text below. Page and Line numbers are referred to clean manuscript file.

---

## [Editor Report · Decision Letter 2]

28 Feb 2022

Utilization of insecticide-treated bed nets among pregnant women in Myanmar – analysis of the 2015-2016 Demographic and Health Survey

PONE-D-22-00065R2

Dear Dr. Aung,

We’re pleased to inform you that your manuscript has been judged scientifically suitable for publication and will be formally accepted for publication once it meets all outstanding technical requirements.

Kind regards,

Myat Htut Nyunt, MMedSc, PhD

Academic Editor

PLOS ONE
---

## [Editor Report · Acceptance letter]

2 Mar 2022

PONE-D-22-00065R2 

Utilization of insecticide-treated bed nets among pregnant women in Myanmar – analysis of the 2015-2016 Demographic and Health Survey 

Dear Dr. Aung:

I'm pleased to inform you that your manuscript has been deemed suitable for publication in PLOS ONE. Congratulations! Your manuscript is now with our production department. 

Kind regards, 

on behalf of

Dr. Myat Htut Nyunt 

Academic Editor

PLOS ONE